biophysics/molecular biology/computer modelling and simulation

pilus, steered molecular dynamics, atomic force microscopy

**Author for correspondence:**
Fabiano Corsetti
e-mail: fabiano.corsetti@gmail.com

# Entropic bonding of the type 1 pilus from experiment and simulation

Fabiano Corsetti[1,2,3,4], Alvaro Alonso-Caballero[4,5], Simon Poly[4,6], Raul Perez-Jimenez[4,7] and Emilio Artacho[4,7,8,9]

[1]Department of Materials, [2]Department of Physics, and [3]The Thomas Young Centre for Theory and Simulation of Materials, Imperial College London, London SW7 2AZ, UK
[4]CIC nanoGUNE, 20018 Donostia-San Sebastián, Spain
[5]Department of Biological Sciences, Columbia University, NY 10027, USA
[6]Chimie et Biologie des Membranes et des Nanoobjets CBMN, Université de Bordeaux, 33600 Pessac, France
[7]Basque Foundation for Science Ikerbasque, 48011 Bilbao, Spain
[8]Theory of Condensed Matter, Cavendish Laboratory, University of Cambridge, Cambridge CB3 0HE, UK
[9]Donostia International Physics Center DIPC, 20018 Donostia-San Sebastián, Spain

 FC, 0000-0002-2275-436X; EA, 0000-0001-9357-1547

The type 1 pilus is a bacterial filament consisting of a long coiled proteic chain of subunits joined together by non-covalent bonding between complementing β-strands. Its strength and structural stability are critical for its anchoring function in uropathogenic *Escherichia coli* bacteria. The pulling and unravelling of the FimG subunit of the pilus was recently studied by atomic force microscopy experiments and steered molecular dynamics simulations (Alonso-Caballero *et al.* 2018 *Nat. Commun.* **9**, 2758. (doi:10.1038/s41467-018-05107-6)). In this work, we perform a quantitative comparison between experiment and simulation, showing a good agreement in the underlying work values for the unfolding. The simulation results are then used to estimate the free energy difference for the detachment of FimG from the complementing strand of the neighbouring subunit in the chain, FimF. Finally, we show that the large free energy difference for the unravelling and detachment of the subunits which leads to the high stability of the chain is entirely entropic in nature.

# 1. Introduction

Over the last two decades, single-molecule force spectroscopy studies using atomic force microscopy (AFM) have provided much

invaluable insight into the mechanical properties of biomolecules [1–5]. In particular, the folding and unfolding for a variety of proteins has been probed in detail.

Such experiments can be complemented from simulation by using atomistic force-fields with steered molecular dynamics [6,7] (SMD) or similar techniques [8,9]. Although simulations can provide additional information of significant interest inaccessible from experiment, it is technically very challenging to quantitatively validate the two against each other [2,5,10,11]. This would be particularly desirable as the limit of accuracy of the force-fields used is generally not known with certainty. Nevertheless, recent studies have given some promising indications of the feasibility of such comparisons [12].

In this study, we attempt to compare AFM experiments and SMD simulations for the pulling and unfolding of the FimG subunit of Gram-negative type 1 bacterial adhesion pili (fimbriae). The mechanical properties of the type 1 pilus are of particular interest because of its key role in initiating infection and keeping the bacteria anchored to the host [13,14]. In our previous work [15], we have characterized the four different subunit types that make up the pilus (FimA, FimF, FimG, FimH), showing that they all share similar properties and an exceptional mechanical stability. Here, instead, we focus in detail on a single subunit (FimG, the middle unit in the tip fibrillum) as a representative example.

As is commonly the case, our main obstacle is that the pulling rates attainable by simulation are many orders of magnitude faster than those used in experiment.[1] Furthermore, the process is far from equilibrium in both cases, as even the results of the much slower experimental pulling appreciably vary by reducing the pulling rate [19]. The raw force–extension traces are therefore not comparable; we focus instead on estimating the change in free energy from the folded to the unfolded state by calculating the distribution of work values for repeated pulling trials, finding a good agreement between experiment and simulation using the mean work estimator. We will discuss the success and limitations of this strategy for our dataset. Finally, we will focus on the physical insights that the simulations provide us beyond the experimental results, not only in terms of the atomistic view of the unfolding process but also in the change of the thermodynamic variables across the transition. In particular, we show that the simulations can help isolate and remove the effect of the artificial linker between subunits needed in the experimental assay, and that the strong bonding between separate subunits holding the pilus together in nature is the sole result of entropic forces.

# 2. Methods

## 2.1. Experimental pulling

AFM force–extension experiments were conducted on a polyprotein sequence composed of the FimG domain of interest at the centre, with two I91 domains on either side used as markers owing to their well-characterized mechanical properties [20] (the set-up is schematically shown in figure 1a). The FimG protein is a single domain, eight-stranded β-barrel which includes the complementing β-strand from FimF, the next protein in the pilus chain. As there is no covalent bonding of this β-strand with the FimG domain, a connecting sequence of four amino acids (DNKQ) was inserted between the two as a flexible linker; this allows the complete unfolding of the protein in the experiment, which would not otherwise be possible after the detachment of the two Fim domains.

The pulling was executed at $400 \, nm \, s^{-1}$, starting with the tip being pushed into the cantilever to produce a force of approximately 1 nN. The pulling was then conducted in the opposite direction up to and beyond the point of polyprotein-to-tip detachment, with the force/extension pair being recorded every 0.2 ms. The spring constant of the cantilever was of approximately $15 \, pN \, nm^{-1}$.

Further details of the experimental set-up as well as of the protein expression and purification can be found in [15].

## 2.2. Data processing of experimental results

The experimental pulling procedure is repeated many times, resulting in a large set of trial traces that need to be analysed. This is particularly important because, as is common in such experiments, the majority of traces are not usable owing to a variety of random unwanted behaviours (too many or too few markers in the sequence, interferences in the unfolding of the proteins, etc.) For the processing of the data, we therefore use an automatic analyser software tool, custom written for this study. It is

---

[1]It should be noted that, apart from the all-atom models discussed here, there are also a variety of more approximate coarse-grained models operating at different levels of detail which have been used to achieve simulated pulling rates closer to experiment [16–18].

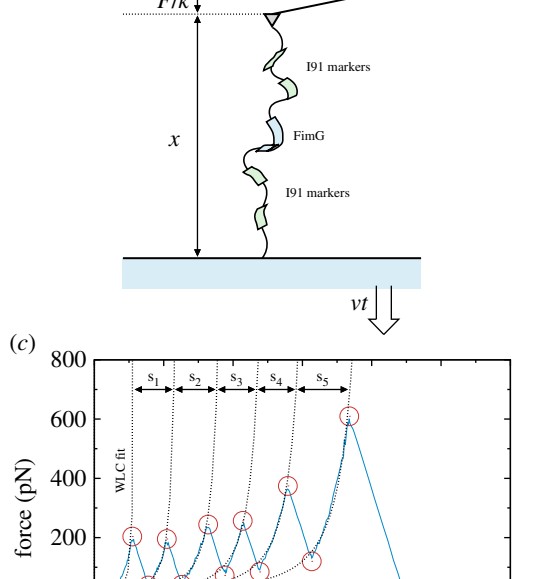

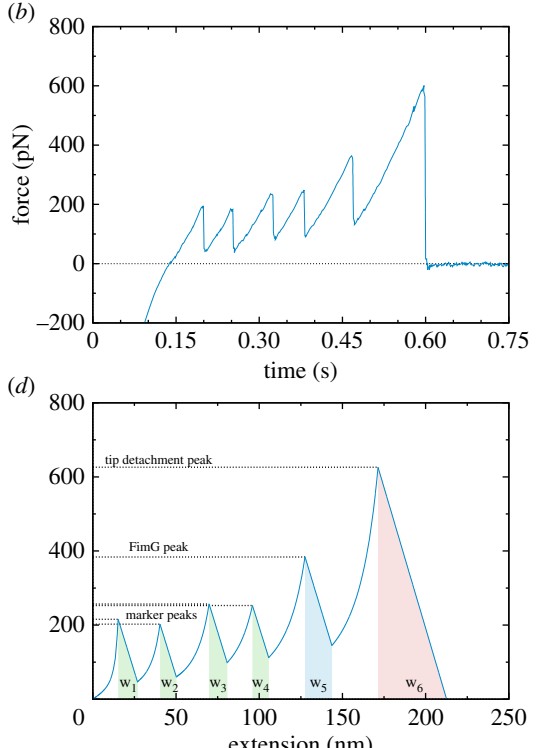

**Figure 1.** Atomic force microscopy (AFM) force-extension experiment. (*a*) Schematic view of the experimental set-up; the FimG protein domain is attached in between two dimers of I91 domains used as markers. The polyprotein sequence is held between a cantilever with spring constant $k$ and a gold surface retracted at speed $v$; the extension $x$ of the whole sequence and the force $F$ on the cantilever are recorded. (*b*) An example force–time trace (after downsampling). (*c*) The corresponding force–extension trace, with the fitted curves from the WLC model for each branch and the automatically determined maxima and minima. The contour length increment $s_i$ owing to the unfolding of each domain is also shown. (*d*) The final force–extension trace using the fitted curves; the shaded areas give the quasi-static work across the transitions $W_i^{\text{tip}}$.

important to note that, although the raw experimental data is shared with Alonso-Caballero *et al.* [15], the analysis presented here is novel and was conducted separately.

The automatic processing is applied to each individual pulling trial, with the following steps:

(i) the force–extension trace is downsampled by averaging data points over a width of 1 nm; this smoothens the trace and eliminates the very fast oscillations that occur especially around the transitions;

(ii) the minima and maxima of the trace (corresponding to the beginning and end points of each branch, respectively) are located (figure 1*c*); this is done by first determining the transition midpoints from the first derivative. A threshold is used to filter out the noise after the tip detachment. The point at which the trace first crosses the force axis is also found; this is used as the origin for further analysis; and

(iii) finally, each branch is independently fitted to a worm-like chain (WLC) model (figure 1*c*). We use the improved interpolation formula by Bouchiat *et al.* [21] to the original by Bustamante *et al.* [22,23], derived for the entropic regime. The formula is given in the electronic supplementary material.

Using this analysis, the trace is either accepted or discarded based on a number of quantitative criteria:

— the presence of a flat, zero-force region at the end of the trace after the final peak, indicating a complete tip detachment; this has to be at least 20 nm in length and with a fitted slope of less than 0.01 pN nm$^{-1}$;

— a reasonable number of extrema, alternating between maxima and minima. The maxima have to be groupable as follows:

(i) 3–5 peaks of less than 300 pN, corresponding to the unfolding of the I91 marker proteins;

(ii) 1 peak of greater than 300 pN, corresponding to the unfolding of the FimG protein;

(iii) 1 peak taller than all previous ones, corresponding to the tip detachment.

— the fitted WLC curves close enough to the real trace (root mean square error $< 100\,\text{pN}$ for all branches); and

— the results of the WLC fits returning reasonable values for the two free parameters, the persistence length $P$ and contour length of the domain $s$:

  (i)  $0.01 < P < 1\,\text{nm}$ for all branches;

  (ii)  $25 < s < 45\,\text{nm}$ for I91, and $s < 100\,\text{nm}$ for FimG.

Some examples of rejected traces are shown in the electronic supplementary material. If all the tests pass, meaning that the trace is accepted for our final analysis set, we can construct a piecewise force–extension trace for the whole pulling experiment using the WLC fit for each branch and the automatically determined maxima and minima (figure 1$d$). From this, we compute the unfolding work value $W^{\text{unfold}}$ for each transition; following Manosas & Ritort [24], this is given by

$$W^{\text{unfold}} = W^{\text{rip}} - \Delta G_s, \tag{2.1}$$

where $\Delta G_s$ is the change in free energy of the whole sequence between the folded and unfolded branches, and $W^{\text{rip}}$ is the quasi-static work across the transition (the area under the trace for the segment connecting the two branches). Therefore, for two branches fit to models $f_1^{\text{WLC}}(x)$ and $f_2^{\text{WLC}}(x)$, and a transition between them from extension $x_1$ to $x_2$, the two terms are calculated as

$$\left. \begin{aligned} W^{\text{rip}} &= \frac{(f_1^{\text{WLC}}(x_1) + f_2^{\text{WLC}}(x_2))(x_2 - x_1)}{2} \\ \text{and} \quad \Delta G_s &= \int_0^{x_2} f_2^{\text{WLC}}(x)\,\mathrm{d}x - \int_0^{x_1} f_1^{\text{WLC}}(x)\,\mathrm{d}x. \end{aligned} \right\} \tag{2.2}$$

## 2.3. Simulated pulling

Molecular dynamics simulations of the unfolding of the FimG domain were carried out using the GROMACS [25] code (version 4.6.5). The protein was set up in the same way as for the experimental polyprotein sequence, with the complementing β-strand from FimF and the connecting DNKQ sequence. The initial model was obtained from a FimC–FimF–FimG–FimH complex previously resolved [26] and made available in the Protein Data Bank (PDB ID: 4J3O) [27]. After separating the subunit with its complementing β-strand, the connecting sequence of amino acids was manually inserted between them to form a single chain (final structure available online; see [28]).

Each individual simulation was carried out with the following steps:

  (i)  the protein is placed at the centre of a box of dimensions $4 \times 4 \times 60\,\text{nm}$, with the pulling axis aligned with the $z$-direction;

  (ii)  the empty space in the box is filled by solvent molecules (3-site water model) placed at random;

  (iii)  some solvent molecules are randomly replaced with monoatomic salt ions to reach an ion concentration of $0.15\,\text{mol}\,\text{l}^{-1}$;

  (iv)  an initial steepest-descent energy minimization is performed to obtain a reasonable starting configuration;

  (v)  the system is equilibrated for 100 ps at 300 K using velocity rescaling with a stochastic term [29]; and

  (vi)  the pulling simulation is performed for 40 ns at 300 K using Nosé–Hoover temperature coupling (no pressure coupling is applied). The pulling is executed at $1\,\text{nm}\,\text{ns}^{-1}$ with a spring constant of $1000\,\text{kJ}\,\text{mol}^{-1}\,\text{nm}^{-2}$; the pulling groups are the nitrogen and carbonyl carbon atoms of the N and C-terminals of the whole protein complex, respectively.

Both the equilibration and pulling simulations use a time step of 1 fs; the pulling force is recorded at every step, while the end-to-end extension of the protein is recorded every 10 steps. All simulations make use of three-dimensional periodic boundary conditions; long-range electrostatics are treated with the smooth particle-mesh Ewald [30] (SPME) method, while van der Waals interactions are calculated with a real-space cutoff of 1.5 nm. The force-fields used are CHARMM27-CMAP [31] for the protein and TIP3P [32] for the solvent.

# 3. Results and discussion

## 3.1. Experimental pulling

The AFM pulling experiment resulted in 4829 individual force–extension traces. After discarding unclear and noisy results (as explained in §2.2), we are left with 186 traces from which to generate statistics (dataset including both accepted and discarded traces available online; see [28]).

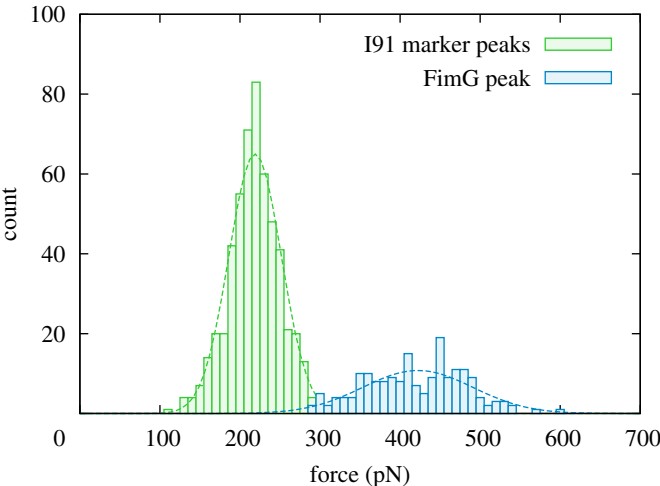

**Figure 2.** Distribution of experimental peak force values. The dashed lines show normal distributions fitted to the data. Distribution of the tip detachment peak not shown.

The aim of using an automatic procedure is to ensure an unbiased and reproducible analysis. The main difficulty in analysing the experimental traces is in differentiating the I91 marker peaks from that of FimG; this is because, although the latter has a mean unfolding force approximately twice that of the former, the distributions are broad enough to result in a small but non-negligible overlap. As the process is irreversible such a distribution of work values is expected and capturing its shape is important for a statistical mechanics analysis.

The relative difference in contour lengths between the I91 and FimG domains is even smaller and is similarly problematic for differentiating the two.[2] Owing to these considerations, our approach is to separate the peaks with a hard barrier at 300 pN, approximately at the minimum overlap point (figure 2). This will inevitably lead to some FimG peaks being misattributed as I91 and the corresponding traces discarded (as there will be no detected FimG peak in these cases). The error introduced in the final analysis is expected to be small.

The analysis in [15] obtains a mean FimG force peak of $431 \pm 4$ pN. This is in good agreement with our automatic procedure, which finds a value of $422 \pm 69$ pN (note that the previous value uses the standard error of the mean as confidence interval, while we use the standard deviation). Similarly, the I91 marker peak, previously independently measured to be $204 \pm 26$ pN [19], is found here to be $219 \pm 33$ pN (both values given with the standard deviation). Our automatic procedure, however, returns somewhat longer domain contour lengths than the accepted values: $48 \pm 11$ nm for FimG and $34 \pm 3$ nm for I91 (previous studies giving $40 \pm 2$ nm [15] and $28.4 \pm 0.3$ nm [19], respectively). It is important to emphasize again here that this discrepancy is not surprising owing to the simplicity of the model, resulting in effective rather than physical contour lengths; for the purposes of energetics we have chosen this approach in order to obtain reproducible and well-fitted curves, and hence better estimates of the unfolding work values.

## 3.2. Simulated pulling

Compared to the experimental pulling, the SMD simulations are performed with a pulling speed which is many orders of magnitude greater, and so a process that is even further out of equilibrium. Furthermore, owing to the computational cost only 10 traces were produced (dataset available online; see [28]). On the other hand, the simulations give us a much more detailed view of the unfolding of the protein with atomic resolution.

The three-dimensional structure of the FimG protein is shown in figure 3a. The structure is held together by two β-sheets on opposite sides, each with three β-strands and one disulfide bond (DB).

[2]In principle they should take two discrete values with no distribution; however, unlike the force peaks, the values are not directly accessible from the experimental measurement but must be inferred from the fitted WLC model. The simplicity of the two-parameter fit means that what we obtain are effective contour lengths which do have a distribution and significant overlap between the two domains. The alternative of fixing the contour length parameter to the expected value for each domain would further limit the fitting freedom and would not help in differentiating the I91 and FimG peaks.

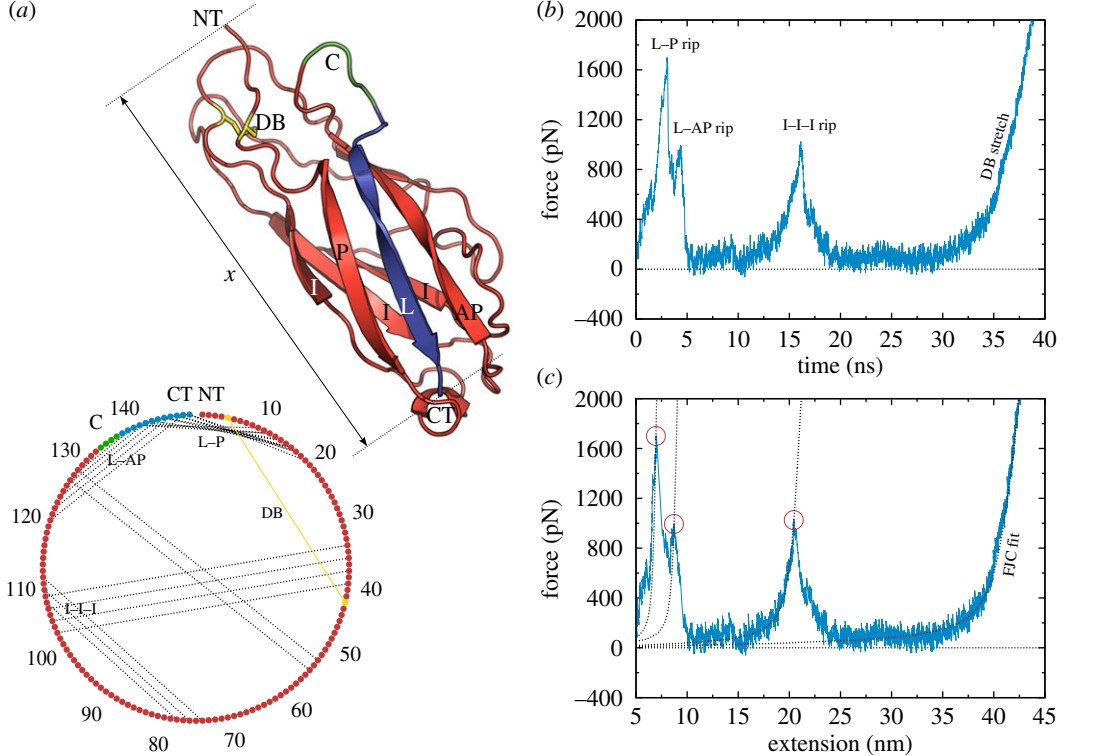

**Figure 3.** Steered molecular dynamics (SMD) simulation. (*a*) Schematic view of the FimG protein (in red) with the linker to the next subunit (L, in blue), connected together by a sequence of four amino acids (C, in green) to form a single chain. The pulling is done from the N and C-terminal amino acids (NT and CT, respectively), and the extension is measured as the distance *x* between them. The lower diagram shows an abstract view of the entire sequence of amino acids, with the HBs between them illustrated by dashed black lines, and the DB as the full yellow line. (*b*) An example force–time trace. (*c*) The corresponding force–extension trace, with the fitted curves from the FJC model for each branch, and the automatically determined maxima. Only the final FJC curve is used in determining the unfolding work.

One of the β-sheets is entirely within the FimG domain, and is made up of an anti-parallel arrangement (the three strands labelled with I). The other sheet is made up from a central strand extending from FimF, the next domain in the pilus chain (labelled with L), hydrogen-bonded (HB) on either side with a strand from within the FimG domain; these two strands are parallel and anti-parallel to the central one (labelled with P and AP, respectively), resulting in a structure reminiscent of a Ψ-loop [33].

The unravelling of the protein occurs over a time-scale of tens of nanoseconds; the simulations are able to resolve the distinct steps involved, as already described in [15]. These are shown in figure 3*b*: first there is a double peak corresponding to the ripping of bonds between the complementing β-strand from the FimF domain and the parallel and anti-parallel strands on either side of it; in all simulations, the first peak is the tallest and corresponds to the ripping off of the parallel strand. Approximately 10 ns later, there is a separate peak, in which the three strands from the intra-protein β-sheet are ripped apart; here, it is harder to distinguish any sub-peaks. At this point, the protein is free to almost completely unravel, except for a loop of 39 amino acids held together by the DB. The DB will not break at the experimental forces [15], and is similarly permanently bonded in the force-field used for the simulation. The simulated pulling therefore ends with the indefinite stretching of this bond.

## 3.3. Comparing free energy differences

As with the experimental traces, we are interested in calculating the work $W^{unfold}$ needed to unfold the protein during the simulation. This is a comparatively much simpler task, as we do not have to worry about excluding traces or distinguishing the marker peaks. Because we know that each simulation corresponds exclusively to the transition of interest, the work is simply obtained by integrating the force value $F$ against the end-to-end extension $x$ (figure 3*c*). However, we must take care to subtract the work of stretching the unravelled protein with the DB. To do so, we fit the freely jointed chain (FJC) model of Smith *et al.* [34] to the last part of the trace; this model also takes into account the

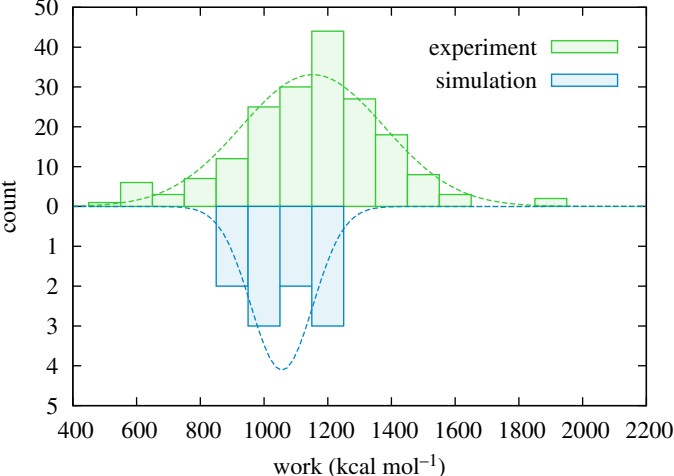

**Figure 4.** Distribution of work values for the unfolding of FimG, $W^{\mathrm{unfold}}$. Note the different scales for experiment and simulation. There are a total of 186 traces from experiment, and 10 traces from simulation. The dashed lines show normal distributions fitted to the data.

elastic modulus of the chain, which is necessary for capturing the non-entropic stretching regime (there are a number of WLC models which also include such a term, but we use the FJC as it provides a better fit to the data; the model formula is given in the electronic supplementary material). The unfolding work as a function of extension is therefore:

$$W^{\mathrm{unfold}}(x) = \int_0^x F(x')\,\mathrm{d}x' - \int_0^x f^{\mathrm{FJC}}(x')\,\mathrm{d}x'. \tag{3.1}$$

Figure 4 shows the distribution of work values obtained from experiment and simulation for the complete unravelling of FimG. There is a striking agreement between the two, with a mean value of $1153 \pm 224\,\mathrm{kcal\,mol^{-1}}$ from experiment, and $1056 \pm 97\,\mathrm{kcal\,mol^{-1}}$ from simulation. This is in contrast to the peak force values, which, as can be seen in figures 1d and 2 for experiment and figure 3 for simulation, differ by a factor of 4.

The large difference in the peak force is as expected from the different pulling rates [35]. Best *et al.* [11] have shown that the unfolding force increases linearly with the logarithm of the pulling speed; our simulated pulling is over 6 orders of magnitude faster than experiment, and the corresponding increase in the peak force is therefore broadly in line with previous studies. Furthermore, the peak force is also affected by the set-up of the whole polyprotein sequence being stretched, which differs between experiment (including the marker proteins) and simulation (only the domain of interest); the work, on the other hand, when properly extracted is independent of these differences.

The approximate equivalence of the work distributions, however, should be considered in more detail. While the reversible work as measured in a quasi-static process (equal to the free energy change across the transition) has to remain constant, our pulling is performed far from equilibrium and the calculated work should therefore have an additional dissipative component. The effect of this dissipative work has been explored in several previous studies [12,36–40] using Jarzynski's equality [41–43].

In highlighting the good agreement of the mean work values, we are effectively making use of the mean work estimator [37] as a way of estimating free energy differences. The Jarzynski estimator is generally considered superior [36,37,39]; however, it is also much more sensitive to the shape of the distribution owing to its exponentially favourable weighting towards low work values. In our case, we are hindered both for simulation, owing to the small sample size, and for experiment, owing to the uncertainties present in the analysis of the results. It seems likely that these uncertainties contribute to a spurious widening of the distribution, as we would expect the true distribution from experiment to be narrower than that from simulation, because the pulling is performed far closer to equilibrium in the former case. This widening will therefore significantly affect the Jarzynski estimator. Concerning the expected bias of the mean work estimator, the following interesting point should be noted, however: most of the dissipation coming from the much larger force needed in the simulations ends up in the work term for the WLC, the unfolding dissipation appearing to be negligible within the statistical noise.

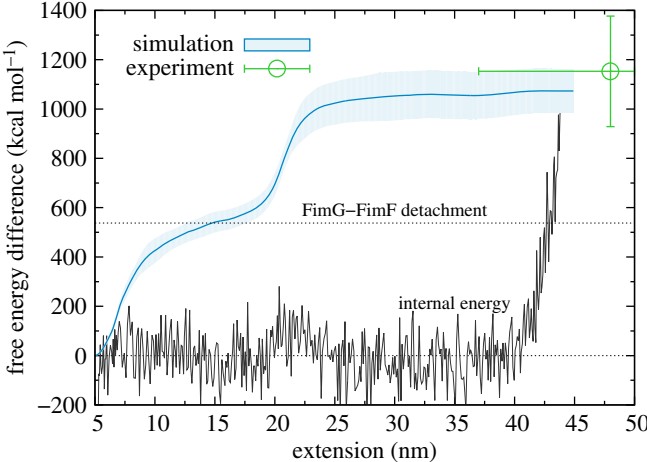

**Figure 5.** Free energy difference for the unfolding using the mean work estimator $\langle W^{\text{unfold}} \rangle$. The error bars for both experiment and simulation are the instantaneous standard deviations from the respective distributions. The internal energy of the system from the simulation is also shown by the full black line. The free energy for detachment between FimG and FimF in simulations without the linker is shown by the dotted horizontal line.

Finally, it is worth pointing out that FimG (along with the other pilus domains) shows exceptional mechanical stability rarely seen in proteins [15]; consequently, the forces and work values are higher than any considered in other studies concerned with non-equilibrium distributions [11,12,24,36–40]. The applicability of previous results to this high-stability regime is therefore a matter worthy of further study.

## 3.4. Physical insights into the unfolding of FimG

As already described in §3.2, the SMD simulations give us detailed insight into the specific steps occurring during the unfolding process. This is of particular interest owing to the fact that the experiments are performed with the artificial linker between FimG and FimF, which is necessary in order to be able to detect the unfolding. From simulation, however, we are able to identify and remove the effect of the linker. This can be seen in figure 5, which plots the free energy increase during the unfolding.

The free energy profile shows an initial sharp increase up to an extension of 10 nm (owing to the L–P and L–AP rips shown in figure 3c), followed by a quasi-plateau with a much shallower increase for around 10 nm, and finally a second sharp increase for around 5 nm (owing to the I–I–I rip). After this, the unfolding is effectively complete and the free energy difference is stable at its final value of almost 1100 kcal mol$^{-1}$.

If the linker were not present there would be a complete detachment of FimG from FimF after the L–AP rip, approximately halving the total free energy difference, and therefore the work needed to break the pilus chain. In order to confirm this, we have carried out additional pulling simulations without the linker; here we can directly measure the free energy difference for detachment, which we find to be 537 ± 34 kcal mol$^{-1}$ (shown by the horizontal dashed line in figure 5). This value is at the same height as the plateau between the ripping of the two β-sheets, confirming our initial assumption that the linker does not interfere with the unfolding mechanism.

It is particularly interesting to analyse the nature of the free energy change across the transition together with the changing patterns in hydrogen bonding. The HBs can be detected at each time step by monitoring the H–O distances both within the protein and between the protein and solvent molecules. The pattern of bonding within the protein before the pulling starts is shown in figure 3a; three-dimensional together with the DB, the HBs are responsible for maintaining the three-dimensional structure of the protein. The unfolding destroys all of these HBs; however, the total number of HBs in the system remains constant across the transition, as, on average, each intra-protein HB is replaced by two new protein–solvent HBs and the breaking of an existing intra-solvent HB.

The constant nature of the bonding is reflected in the internal energy of the system, plotted in figure 5 (the reference is taken to be the average internal energy of the system before pulling). The internal energy does not change across the transition, as the overall number of bonds remains constant. The increase that can be seen at the end (for extensions greater than 40 nm) corresponds to the stretching of the DB.

The free energy difference, therefore, can be attributed entirely to a decrease in entropy. Here we do not refer to the entropic force required to stretch the protein (this contribution is subtracted with the FJC term of equation (3.1)); rather, it is a change in the configurational entropy of the solvent in switching from creating HBs with itself to bonding to the free β-strands of the protein. There are around 40 intra-protein HBs in the folded protein, resulting in an average entropic cost of approximately 30 kcal mol$^{-1}$ for breaking each one. Interestingly, this is an order of magnitude higher than the reported entropic cost of transferring a water molecule to a protein cavity [44], and with no balancing enthalpic contribution.

More broadly, the simulations clearly demonstrate the entropic nature of β-sheet bonding in the FimG protein, and the very high mechanical stability that can be obtained from entropy alone.

# 4. Conclusion

Through the mechanism of β-strand complementation, the type 1 pilus succeeds in creating a highly stable chain of thousands of Fim subunits with no covalent bonding between them, capable of withstanding significant force and therefore serving as an effective anchoring device for bacteria [45]. Experimental force spectroscopy can provide important information about the mechanical stability of the chain; however, the use of atomistic simulations complements these findings with insight into the nature of the bonding. This is of particular interest as the role of hydrophobic interactions and HBs in the stabilization of β-sheets is a topic of ongoing research [46,47]. Our simulations show that entropic contributions are solely responsible for the strong supramolecular bonding that holds the chain together, as well as most of the structural bonding within a single subunit (with the exception of the covalent DB). The simulations, in turn, are supported by a careful quantitative comparison with experiment; despite the huge difference in pulling speeds and peak force values, we show that processing the force-extension data to obtain work values results in overlapping distributions. This goes some way towards validating the accuracy of the force-fields used. Nevertheless, it is important to acknowledge the limits of the comparison, and in particular the unclear effect of the dissipative work component. We suggest, therefore, that future work on non-equilibrium distributions should focus on processes with high work values as in the present example, for which little data are available at present.

Data accessibility. Datasets are available at the Dryad Digital Repository: http://dx.doi.org/10.5061/dryad.573n5tb44 [28].
Authors' contributions. S.P., A.A.C. and R.P.J. performed the AFM experiments. F.C. performed the SMD simulations and the analysis of the experimental results. F.C. and E.A. analysed the data and interpreted the results. F.C. wrote the manuscript; all authors edited the manuscript.
Competing interests. We declare we have no competing interests.
Funding. This work was partly funded by grant nos. FIS2012-37549-C05, FIS2015-64886-C5-1-P, BIO2016-77390-R and BFU2015-71964 from the Spanish Ministry of Economy and Competitiveness (MINECO), and Exp. 97/14 (Wet Nanoscopy) from the Programa Red Guipuzcoana de Ciencia, Tecnología e Innovación, Diputación Foral de Gipuzkoa. A.A.C. was funded by the predoctoral programme of the Basque Government. R.P.J. was supported by the CIG-Marie Curie Reintegration programme FP7-PEOPLE-2014 from the European Commission. The calculations were performed on the arina HPC cluster (Universidad del País Vasco/Euskal Herriko Unibertsitatea, Spain). SGIker (UPV/EHU, MICINN, GV/EJ, ERDF and ESF) support is gratefully acknowledged.
Acknowledgements. We thank Felix Ritort and Ronen Zangi for useful discussions.

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
