## [Reviewer comments · Royal Society Open Science]

Review History

RSOS-200183.R0 (Original submission)

Review form: Reviewer 1

Is the manuscript scientifically sound in its present form?

Yes

Are the interpretations and conclusions justified by the results?

Yes

Is the language acceptable?

Yes

Do you have any ethical concerns with this paper?

No

Have you any concerns about statistical analyses in this paper?

No

Recommendation?

Accept with minor revision (please list in comments)

Comments to the Author(s)

Referee report on RSOS-200813

Entropic bonding of the type 1 pili from experiment and simulation by F. Corsetti, A. Alonso-Canallero, S. Poly, R. Perez-Jimenez, and E. Artacho

This article reports a combined experimental and computational study of the pulling of a particular biomolecule, the FimG subunit of the tip fibrillum of Gram-negative type 1 bacterial adhesion pili. This work follows a recent paper published by the same authors, but using here more systematic analysis tools for the experimental data and a deeper treatment of the numerical data.

The measurements involve pulling and unfolding by AFM, with markers added on both ends of the biomolecule to distinguish its unfolding from that in FimG. Here an automated procedure was used to select the data compatible for analysis (a minority of them) but the authors convincingly justify the need for such an automation.

The simulations are also clearly explained, but the entropic nature of the unfolding transition could be discussed a bit further. I understand that with a nearly constant internal energy along the pulling pathway the transition has to be entropic in nature, but the explanation for this constant internal energy (based on the hydrogen bonds made with the solvent) perhaps a bit weak, and I recommend the authors to illustrate their claim by showing the numbers of HB.

Further minor comments:

-to help the reader, perhaps provide in appendix the expressions for the WLC and FJC forms used in the data analysis

-the availability of all data (used and not used) is appreciated, but perhaps in appendix the authors could also show typical examples of rejected traces

To conclude, I think this is an interesting paper showing how experiment and simulation can work together to bring additional information about the physical chemistry of complex systems. I recommend it for publication in Royal Society Open Science.

Review form: Reviewer 2 (Debayan Chakraborty)

Is the manuscript scientifically sound in its present form?

Yes

Are the interpretations and conclusions justified by the results?

Yes

Is the language acceptable?

Yes

Do you have any ethical concerns with this paper?

No

Have you any concerns about statistical analyses in this paper?

No

Recommendation?

Accept with minor revision (please list in comments)

Comments to the Author(s)

The authors present a nice account of the entropic origin of the forces holding the different subunits of the Gram Negative I bacterial adhesion pili using experiment and computer simulations. The authors need to address some minor points, before publication can be recommended:

(1) In the pulling experiment, I91 markers are attached to the FilmG domain. However, the simulations do not employ any surrogates for the I91 markers, and the pulling forces are applied directly to the two termini of the protein complex. The authors should describe in more detail how these differences in the setup may complicate a one-to-one comparison between experiment and simulations.

(2) As rightly noted by the authors, different pulling rates may cause large deviations in the peak force. Besides the paper by Best et al., the authors should also cite the work of Lee and Thirumalai (Biophys J., 86, 2641, 2004), which I believe is quite relevant in this context.

(3) Besides, atomically detailed simulations, coarse-grained simulations at different resolutions also complement AFM experiments, and provide critical insight. I think in the interest of the reader, it is worth including some relevant references.

Decision letter (RSOS-200183.R0)

10-Mar-2020

Dear Dr Corsetti

On behalf of the Editors, I am pleased to inform you that your Manuscript RSOS-200183 entitled "Entropic bonding of the type 1 pilus from experiment and simulation" has been accepted for publication in Royal Society Open Science subject to minor revision in accordance with the referee suggestions. Please find the referees' comments at the end of this email.

The reviewers and handling editors have recommended publication, but also suggest some minor revisions to your manuscript. Therefore, I invite you to respond to the comments and revise your manuscript.

- Ethics statement

- Data accessibility

It is a condition of publication that all supporting data are made available either as supplementary information or preferably in a suitable permanent repository. The data accessibility section should state where the article's supporting data can be accessed. This section should also include details, where possible of where to access other relevant research materials such as statistical tools, protocols, software etc can be accessed. If the data has been deposited in an external repository this section should list the database, accession number and link to the DOI

for all data from the article that has been made publicly available. Data sets that have been deposited in an external repository and have a DOI should also be appropriately cited in the manuscript and included in the reference list.

<http://datadryad.org/submit?journalID=RSOS&manu=RSOS-200183>

- **Competing interests**

- **Authors' contributions**

- **Acknowledgements**

- **Funding statement**

Because the schedule for publication is very tight, it is a condition of publication that you submit the revised version of your manuscript before 19-Mar-2020. Please note that the revision deadline will expire at 00.00am on this date. If you do not think you will be able to meet this date please let me know immediately.

When submitting your revised manuscript, you will be able to respond to the comments made by the referees and upload a file "Response to Referees" in "Section 6 - File Upload". You can use this to document any changes you make to the original manuscript. In order to expedite the

processing of the revised manuscript, please be as specific as possible in your response to the referees. We strongly recommend uploading two versions of your revised manuscript:

If your manuscript is newly submitted and subsequently accepted for publication, you will be asked to pay the article processing charge, unless you request a waiver and this is approved by Royal Society Publishing. You can find out more about the charges at <https://royalsocietypublishing.org/rsos/charges>. Should you have any queries, please contact openscience@royalsociety.org.

on behalf of Dr David Wales (Associate Editor) and Pietro Cicuta (Subject Editor)

Associate Editor Comments to Author (Dr David Wales):

Associate Editor: 1

Comments to the Author:

Please revise your manuscript in accord with the suggestions of the two referees, and detail the changes.

Reviewer comments to Author:

Reviewer: 1

Comments to the Author(s)

Referee report on RSOS-200813

Entropic bonding of the type 1 pilus from experiment and simulation by F. Corsetti, A. Alonso-Canallero, S. Poly, R. Perez-Jimenez, and E. Artacho

This article reports a combined experimental and computational study of the pulling of a particular biomolecule, the FimG subunit of the tip fibrillum of Gram-negative type 1 bacterial adhesion pili. This work follows a recent paper published by the same authors, but using here more systematic analysis tools for the experimental data and a deeper treatment of the numerical data.

The measurements involve pulling and unfolding by AFM, with markers added on both ends of the biomolecule to distinguish its unfolding from that in FimG. Here an automated procedure was used to select the data compatible for analysis (a minority of them) but the authors convincingly justify the need for such an automation.

The simulations are also clearly explained, but the entropic nature of the unfolding transition could be discussed a bit further. I understand that with a nearly constant internal energy along the pulling pathway the transition has to be entropic in nature, but the explanation for this constant internal energy (based on the hydrogen bonds made with the solvent) perhaps a bit weak, and I recommend the authors to illustrate their claim by showing the numbers of HB.

Further minor comments:

-to help the reader, perhaps provide in appendix the expressions for the WLC and FJC forms used in the data analysis

-the availability of all data (used and not used) is appreciated, but perhaps in appendix the authors could also show typical examples of rejected traces

To conclude, I think this is an interesting paper showing how experiment and simulation can work together to bring additional information about the physical chemistry of complex systems. I recommend it for publication in Royal Society Open Science.

Reviewer: 2

Comments to the Author(s)

The authors present a nice account of the entropic origin of the forces holding the different subunits of the Gram Negative I bacterial adhesion pili using experiment and computer simulations. The authors need to address some minor points, before publication can be recommended:

(1) In the pulling experiment, I91 markers are attached to the FilmG domain. However, the simulations do not employ any surrogates for the I91 markers, and the pulling forces are applied directly to the two termini of the protein complex. The authors should describe in more detail how these differences in the setup may complicate a one-to-one comparison between experiment and simulations.

(2) As rightly noted by the authors, different pulling rates may cause large deviations in the peak force. Besides the paper by Best et al., the authors should also cite the work of Lee and Thirumalai (Biophys J., 86, 2641, 2004), which I believe is quite relevant in this context.

(3) Besides, atomically detailed simulations, coarse-grained simulations at different resolutions also complement AFM experiments, and provide critical insight. I think in the interest of the reader, it is worth including some relevant references.

Author's Response to Decision Letter for (RSOS-200183.R0)

See Appendix A.

Decision letter (RSOS-200183.R1)

19-Mar-2020

Dear Dr Corsetti,

It is a pleasure to accept your manuscript entitled "Entropic bonding of the type 1 pilus from experiment and simulation" in its current form for publication in Royal Society Open Science. The comments of the reviewer(s) who reviewed your manuscript are included at the foot of this letter.

on behalf of Dr David Wales (Associate Editor) and Pietro Cicuta (Subject Editor)
openscience@royalsociety.org

Appendix A

We would like to thank the reviewers for their careful reading of our article and their helpful comments. We are very pleased with their assessment that the research is interesting and clearly explained, and suitable for publication in Royal Society Open Science.

We have carefully considered all of the reviewers' suggestions and have made several modifications and additions to the manuscript in order to address them; we feel that the article has improved as a result. Below we give a point-by-point response for each comment, and provide a list of changes for the whole manuscript.

List of changes

- Added footnote 1 in the Introduction (Sec. 1, page 1, column 2) and Refs. 16-18.
- Added a Supplementary Material document, with references to it where appropriate in the main text (two references in Sec. II B, page 2, column 2; one reference in Sec. III C, page 6, column 1).
- Added Ref. 35 in Sec. III C, page 6, column 1.
- Added a discussion of how the peak force is affected by the experimental setup in Sec. III C, page 6, column 2 (first paragraph).
- Expanded the discussion of the entropic nature of the bonding, Sec. III D, page 7, column 2 (first and third paragraphs), including the addition of Ref. 44.

Reviewer: 1

This article reports a combined experimental and computational study of the pulling of a particular biomolecule, the FimG subunit of the tip fibrillum of Gram-negative type 1 bacterial adhesion pili. This work follows a recent paper published by the same authors, but using here more systematic analysis tools for the experimental data and a deeper treatment of the numerical data.

The measurements involve pulling and unfolding by AFM, with markers added on both ends of the biomolecule to distinguish its unfolding from that in FimG. Here an automated procedure was used to select the data compatible for analysis (a minority of them) but the authors convincingly justify the need for such an automation.

The simulations are also clearly explained, but the entropic nature of the unfolding transition could be discussed a bit further. I understand that with a nearly constant internal energy along the pulling pathway the transition has to be entropic in nature, but the explanation for this constant internal energy (based on the hydrogen bonds made with the solvent) perhaps a bit weak, and I recommend the authors to illustrate their claim by showing the numbers of HB.

We agree with the reviewer that this is an important part of the results. We have therefore worked to improve and expand the discussion in Sec. III D, with the addition of this new text and reference:

'[...] The unfolding destroys all of these HBs; however, the total number of HBs in the system remains constant across the transition, as, on average, each intra-protein HB is replaced by

two new protein–solvent HBs and the breaking of an existing intra-solvent HB. [...] The free energy difference, therefore, can be attributed entirely to a decrease in entropy. Here we do not refer to the entropic force required to stretch the protein (this contribution is subtracted with the FJC term of Eq. 3); rather, it is a change in the configurational entropy of the solvent in switching from creating HBs with itself to bonding to the free β -strands of the protein. There are ~ 40 intra-protein HBs in the folded protein, resulting in an average entropic cost of ~ 30 kcal mol⁻¹ for breaking each one. Interestingly, this is an order of magnitude higher than the reported entropic cost of transferring a water molecule to a protein cavity [44], and with no balancing enthalpic contribution.'

Although we understand the reviewer's suggestion to show the number of HBs, we feel that this would add unnecessary complication to the results due to the well-known ambiguities in arriving at a rigorous geometric or energetic definition of the HB which can be used for such an analysis. We therefore prefer to leave the very clear and unambiguous demonstration of the entropic nature of the bonding given by Fig. 5, and maintain a more qualitative description of the HBs in the discussion.

Further minor comments:

-to help the reader, perhaps provide in appendix the expressions for the WLC and FJC forms used in the data analysis

We have added a new Supplementary Material document which gives the expressions for the WLC and FJC used in the analysis, with the reference for each (Eqs. S1 and S2). The reader is referred to this document in the main text when the WLC and FJC are introduced.

-the availability of all data (used and not used) is appreciated, but perhaps in appendix the authors could also show typical examples of rejected traces

We thank the reviewer for this interesting suggestion; we agree that showing typical examples of rejected traces is very useful for explaining our automatic analysis procedure. We have added four such examples in the new Supplementary Material document (Fig. S1), illustrating four different possible reasons for rejection. The reader is referred to this document in the main text when the rejection criteria are explained.

To conclude, I think this is an interesting paper showing how experiment and simulation can work together to bring additional information about the physical chemistry of complex systems. I recommend it for publication in Royal Society Open Science.

Reviewer: 2

The authors present a nice account of the entropic origin of the forces holding the different subunits of the Gram Negative I bacterial adhesion pili using experiment and computer simulations. The authors need to address some minor points, before publication can be recommended:

(1) In the pulling experiment, 191 markers are attached to the FilmG domain. However, the simulations do not employ any surrogates for the 191 markers, and the pulling forces are applied directly to the two termini of the protein complex. The authors should describe in more detail how these differences in the setup may complicate a one-to-one comparison between experiment and simulations.

We thank the reviewer for this insightful point; it is true that the addition of the markers has an effect on the peak force, which is another reason (on top of the different pulling speeds) why we discourage a direct comparison of the peak force between simulation and experiment. However, the work obtained by integrating the force-extension curve as described in our article is independent of these differences, and so is a valid comparison to attempt.

We agree with the reviewer that this is an important and interesting point to clarify. We have now added a sentence explaining this in Sec. III C: *'Furthermore, the peak force is also affected by the setup of the whole polyprotein sequence being stretched, which differs between experiment (including the marker proteins) and simulation (only the domain of interest); the work, on the other hand, when properly extracted is independent of these differences.'*

(2) As rightly noted by the authors, different pulling rates may cause large deviations in the peak force. Besides the paper by Best et al., the authors should also cite the work of Lee and Thirumalai (Biophys J., 86, 2641, 2004), which I believe is quite relevant in this context.

We thank the reviewer for bringing this paper to our attention; we agree that it is appropriate to cite when discussing how the peak force is affected by the pulling speed. We have now done so in Sec. III C.

(3) Besides, atomically detailed simulations, coarse-grained simulations at different resolutions also complement AFM experiments, and provide critical insight. I think in the interest of the reader, it is worth including some relevant references.

We agree that discussing coarse-grained methods as well as all-atom ones adds some more context to our study. We have done so with the addition of a footnote in the Introduction, and the inclusion of three new references: *'It should be noted that, apart from the all-atom models discussed here, there are also a variety of more approximate coarse-grained models operating at different levels of detail which have been used to achieve simulated pulling rates closer to experiment [16– 18].'*